# A Quantitative Exploration of Reconciliation: Evidence from Colombia

Eliana Sanandres Campis [1,*], Ivonne Molinares-Guerrero [1], Roberto González Arana [1] and Melissa Martínez Pérez [2]

1   Division of Humanities and Social Sciences, Department of History and Social Sciences, Universidad del Norte, Barranquilla 080001, Colombia
2   Social Development, Universidad del Norte, Barranquilla 080001, Colombia
*   Correspondence: esanandres@uninorte.edu.co

**Abstract:** The reconciliation of societies in negotiated transitions from civil war to peace represents a practical challenge. While the political dimension concerns the construction of socio-political relations, the interpersonal dimension focuses on intergroup relationships. Empirical evidence shows that reconciliation should not assign primacy to one dimension over another; rather, it should address the interaction between them. However, research on this topic is scarce. There is a need to develop an instrument to assess the political and interpersonal dimensions of reconciliation in peacebuilding contexts. This study developed the Political and Interpersonal Reconciliation Scale (PIRS) and assessed its psychometric properties based on a sample of Colombian population after the peace agreement between the Colombian government and the FARC guerrilla group. The results show the validity of a factorial structure for two components as well as an acceptable Cronbach's alpha. Concerning external validity, in line with the existing literature, the scale under study was positively related to confidence in the peace agreement, trust in the ex-combatants, willingness to share with the adversary and community identification. This study provides evidence that the Political and Interpersonal Reconciliation Scale is a valid and reliable instrument for evaluating reconciliation in peacebuilding contexts.

**Keywords:** reconciliation; peacebuilding; conflict; measurement; Colombia

## 1. Introduction

Reconciliation is the main component of peacebuilding processes (Lederach 1997). It goes beyond conflict resolution's formal agenda insofar as it seeks to change the attitudes, beliefs, motivations, and nature of the relationships between the actors in conflict (Bar-Tal and Bennink 2004). This political and relational process seeks to guarantee the conditions that allow a more acceptable development in terms of the way in which two parties relate (Nordquist 2007).

Recently, there has been a lack of consensus concerning the variables associated with the study of reconciliation. Some approaches group them into five categories: psychosocial recovery, rapprochement between the parties in conflict, intergroup resignification, emotions, and conflict management (Alzate and Dono 2017). However, these variables predominantly entail a psychological content that may lead to an 'over-psychologized' vision of the concept (Rouhana 2011, p. 300). Other approaches focus on reconciliation as a political process that must guarantee the necessary conditions for rebuilding social relationships (Bar-Tal and Bennink 2004; Bloomfield et al. 2003; Nordquist 2007), while some of them are more oriented toward pragmatism and problem-solving mediation (Princen 1992). However, the distinction between these approaches is not always clearcut (Shamir and Shikaki 2002). This article argues that a better comprehension of reconciliation is made possible by considering two dimensions simultaneously: political and interpersonal.

The political dimension refers to the institutional framework for reconciliation, which focuses on the construction of socio-political relationships when a conflict ends (Bloomfield 2006). According to Rosoux (2009), this dimension is based on a structuralist approach that prioritizes institutional relationships of cooperation and political interdependence in which the parties in conflict establish mutually accepted mechanisms to reduce the perception of threat and resolve their differences. These mechanisms vary according to the context and needs of the society that seeks their implementation, including justice reforms and victim reparation programs (Afzali and Colleton 2003; Firchow 2017; Joshi and Melander 2017; Waardt and Weber 2019).

On the other hand, the interpersonal dimension focuses on the construction of intergroup relationships and conceives reconciliation as a process that requires the implementation of initiatives at the community level, where commonly shared beliefs, attitudes, motivations, and emotions are developed. This dimension deals with the interaction between individuals who have each delegitimized the other based on a conflict and must therefore redefine the terms of their future coexistence (Sanandres and Molinares 2020). In this sense, reconciliation processes must recognize social support networks and community participation as important mechanisms for the recovery that follows traumatic situations. These can facilitate the restoration of fractured social relationships in the context of political violence, where there is generally an emotional climate of fear, social isolation, and mistrust (Schoof et al. 2018).

Research on reconciliation shows a clear dichotomy between the political dimension ('from the top down') and interpersonal dimension ('from the bottom up') (Bloomfield 2006, p. 25). In this sense, Wilmer (1998) suggests that even though the political dimension's structural measures are crucial for establishing an institutional framework to guarantee the minimum level of trust between contending parties, they are inadequate for rebuilding social relations at the community level which has been fractured by the conflict. In contrast, Van Der Merwe et al. (2009) assert that direct interaction between people is neither effective nor sustainable if no institutional framework exists to support it. However, their complementarity suggests a fundamental interaction. Although the reconciliation process can begin with either political leaders or community bases, to be effective. 'It must always proceed from the top down and bottom up simultaneously' (Bar-Tal and Bennink 2004, p. 27). In addition, empirical evidence shows that a reconciliation process should not assign primacy to one dimension over another; rather, it should address the interaction between them, emphasizing their complementarity. However, some of the reconciliation literature still addresses the political and the interpersonal dimensions separately (Maoz 2011; Siani-Davies and Katsikas 2009), and it is surprising that both dimensions rarely enter simultaneously into this analysis.

As a changing phenomenon and practice, it is important to measure the psychometric properties of reconciliation in its political and interpersonal dimensions in order to identify the specific combination of social factors that make it possible. As Rettberg and Ugarriza (2016) point out, when reconciliation means different things for society and for policymakers, such a goal may be more difficult to attain, and social conflicts may persist over time. In the absence of a proper characterization, policymakers may be at a loss as to how to fulfill promises and expectations related to future reconciliation, while society may feel that its fundamental needs are being neither heard nor addressed.

Some historical instances where policymakers have benefited from reconciliation psychometric scales have taken place in South Africa, Cyprus, Rwanda and Colombia. In South Africa, the Reconciliation Barometer (Institute for Justice and Reconciliation 2021) provides a representative measure of citizen's attitudes to national reconciliation, social cohesion, transformation and democratic governances. In Cyprus, The Center for Sustainable Peace and Democratic Development (2013) developed the Social Cohesion and Reconciliation (SCORE) Index to measure the impact of peacebuilding activities and help the general public, policymakers and practitioners understand how social factors interact with one another and how they influence the process of social cohesion and

reconciliation. In Rwanda, the Reconciliation Barometer (National Unity and Reconciliation Commission of Rwanda 2020) has been used since 2016 to assess the status of reconciliation through citizens' views and experiences, identify the reconciliation favorable factors and challenges, and suggest recommendations for a way forward. In the case of Colombia, the Reconciliation Barometer came one year after the 2016 peace agreement between the government and the FARC-EP guerrilla group and sought to investigate the factors that have a positive or negative impact on reconciliation at the national, regional or municipal level (Programa de Alianzas para la Reconciliación 2017). These experiences note that reconciliation presents complexities due to the measurement at multiple levels of analysis and across dimensions that may not synchronize.

The present study focuses on the complementarity between the political and interpersonal dimensions of reconciliation. The aim is to develop the Political and Interpersonal Reconciliation Scale and assess its psychometric properties during a peacebuilding process. The specific objectives are to examine its structural validity and reliability, and to assess its validity in terms of the relationship to four external variables: confidence in the peace agreement, trust in the ex-combatants, willingness to share with the adversary, and community identification. The hypothesis is that the Political and Interpersonal Reconciliation Scale is positively associated with these variables as suggested by the literature. However, as the political and interpersonal dimensions of reconciliation respond to different determinants, it is expected to find associations in different ways. For example, the political dimension is strongly associated with confidence in the peace agreement (Bloomfield et al. 2003; Wolpe and McDonald 2008), while the interpersonal dimension is strongly associated with community identification (Bloomfield 2006) and willingness to trust in the ex-combatants (Alzate et al. 2009; Wagner 2006). Both dimensions are also expected to be associated with willingness to share with the adversary in a new scenario of peace (Alzate et al. 2009).

The context of the current study is the peace process in Colombia, which represents one of the most complex, challenging processes due to unresolved grievances and the presence/permanence of armed groups. In Colombia, generations have passed without experiencing a sense of peace in the nation. Since the 19th century, Colombians have experienced ample civil wars and military truces such as the Bolsheviks of Lebanon (1929) and the Bogotazo (1948), as well as more than fifteen internal armed confrontations that culminated in the Thousand Days War (Celestina 2018). Although it has made peace agreements and amnesties with armed groups, these have been mostly unsuccessful. The last cycle of violence that gave rise to the internal armed conflict emerged in the early 1960s with the formation of the guerrillas Fuerzas Armadas Revolucionarias de Colombia—Ejército del Pueblo (FARC-EP), Ejército de Liberación Nacional (ELN), Ejército Popular de Liberación (EPL), and later, the Movimiento 19 de abril (M-19).

In October 2012, a table for new peace talks was established in Norway between the Colombian government and FARC-EP guerrillas, and the negotiation materialized in 2016 with the signing of the General Agreement for the Termination of the Conflict and the Construction of a Stable and Lasting Peace. At that time, a complex peacebuilding process had begun, including the enforcement of a legal framework at international and national levels. This encouraged reconciliation to be spoken of in terms of being a project for the nation rather than as a process concerning the victims.

However, reconciliation has been incredibly complex because of multiple factors, such as a scenario of polarization where a large political sector of the country does not agree with the terms of the agreement and is exerting multiple pressures against it as a result. Obvious disputes remain as well as a tough battle to construct a memory that makes visible the perpetrators and those responsible for the violence. In this context, reconciliation is a structural, fundamental part of the post-agreement phase, and it is necessary to understand it as an opportunity for beginning to deconstruct, measure, and define what post-agreement entails.

## 2. Materials and Methods

### 2.1. Participants

We used a convenience sampling strategy to choose a sample of university students who were easily accessible and cooperative. These students were located in the north coast of Colombia, one of the most affected regions of the conflict. To contact them, we developed an open workshop about reconciliation for undergraduate students and presented the purpose of the study. A total of 171 students who were enrolled in several undergraduate courses volunteered to participate. Of this sample, 54.4% were men, 43.3% were women, and 2.3% members belonged to the LGBTI (lesbian, gay, bisexual, transgender, and intersex) community. The mean age recorded was twenty years (SD = 4.2). Table 1 presents descriptive data of the sample's sociodemographic variables. Regarding the sample's sociodemographic characteristics, it should be noted that in terms of percentages, there was a balance of self-identification by gender and self-recognition of participants' political ideology, including 25% proportions in the politically left, center, and right categories as well as a remaining 25% that did not identify with any of them.

**Table 1.** Descriptive data of the sample's sociodemographic variables.

|  |  | Men (n = 93) | Women (n = 74) | LGBTIQ+ (n = 4) |
|---|---|---|---|---|
| Marital status |  |  |  |  |
|  | Single | 90 (97%) | 72 (97%) | 3 (75%) |
|  | Married | 1 (1%) | 1 (1%) |  |
|  | Cohabiting | 2 (2%) | 1 (1%) | 1 (25%) |
| Social status |  |  |  |  |
|  | High | 19 (21%) | 18 (25%) |  |
|  | Middle | 51 (55%) | 33 (45%) | 2 (50%) |
|  | Low | 21 (21%) | 21 (29%) | 2 (50%) |
|  | No answer | 2 (2%) | 2 (3%) |  |
| Ethnic origin |  |  |  |  |
|  | Multiethnic | 44 (47% | 36 (49%) | 3 (75%) |
|  | White | 23 (25%) | 16 (22%) |  |
|  | Black | 6 (6%) | 6 (8%) |  |
|  | Indigenous | 3 (3%) | 1 (1%) |  |
|  | Other | 17 (18%) | 15 (20%) | 1 (25%) |
| Religion |  |  |  |  |
|  | Catholic | 55 (59%) | 31 (42%) | 2 (50%) |
|  | Christian | 20 (22%) | 18 (24%) |  |
|  | Jewish | 1 (1%) |  |  |
|  | None | 7 (8%) | 14 (19%) | 2 (50%) |
|  | No answer | 10 (10%) | 11 (14%) |  |
| Political ideology |  |  |  |  |
|  | Left | 27 (29% | 17 (23%) | 1 (25%) |
|  | Center | 22 (24%) | 25 (34%) | 1 (25%) |
|  | Right | 24 (26%) | 17 (23%) |  |
|  | No answer | 20 (22%) | 15 (20%) | 2 (50%) |
| Victims |  |  |  |  |
|  | Yes | 24 (25%) | 22 (30%) | 1 (25%) |
|  | No | 51 (55%) | 42 (57%) | 2 (50%) |
|  | No answer | 18 (20%) | 10 (13%) | 1 (25%) |

Note. Numbers and percentages of sociodemographic variables.

### 2.2. Instrument

For this study, we developed the Political and Interpersonal Reconciliation Scale which contains twelve items from studies validated in peacebuilding contexts (Alzate et al. 2013; Mukashema and Mullet 2010; Taylor 2015). Questionnaires were administered individually by two trained research assistants in online sessions. The scale took approximately 20 min to complete.

The political dimension included six items that evaluated people's attitudes toward the implementation of justice and reparation mechanisms at the end of the conflict. The items correspond to a five-point Likert-type rating scale, where '1' represents complete agreement and '5' stands for complete disagreement. In this way, the scale assesses people's degree of conformity by utilizing the following statements: 'Families, houses, infrastructure and communities affected during the conflict must be rebuilt', 'those affected during the conflict should have a greater voice and participation in political discussions', 'it is important to reduce penalties conditioned by reconstruction and reparation tasks within the law of justice and peace', 'amnesty is a fair result if those responsible for violations tell the truth about what they have done', 'amnesty is a fair result if those responsible for the violations acknowledge the damage they caused', and 'amnesty is a fair result if those responsible for the violations express apologies or ask forgiveness'.

The interpersonal dimension included six items that appraised the extent to which people perceived that their social relationships were a source of support, and their community was a safe space to participate in encounters with others. These items also correspond to a five-point Likert-type rating scale, as described above, but they measured an individual's degree of conformity via the following statements: 'I have someone with whom I can share my greatest concerns and fears', 'I have someone to turn to for suggestions on how to handle a personal problem', 'I have someone who understands my problems', 'I collaborate in organizations and associations in my neighborhood or community', 'I participate in social activities in my neighborhood or community', and 'I participate in petitions that are circulated in my neighborhood or community'.

For an external validation, the study included four variables that were conceptually related to reconciliation's political and interpersonal dimensions. According to the literature, in societies where the parties in conflict are negotiating the transition from civil war to peace, the confidence in a peace agreement is expected to be related to the political dimension of reconciliation (Bloomfield et al. 2003; Wolpe and McDonald 2008). Meanwhile, the identification of individuals with their community and the willingness to trust the ex-combatants are expected to be related to the interpersonal dimension of reconciliation (Alzate et al. 2009; Bloomfield 2006; Wagner 2006). Both dimensions are also expected to be related to a willingness to share with those who caused harm during the conflict in a new scenario of peace.

In terms of measuring confidence in the peace agreement, trust in ex-combatants, community identification, and willingness to share with the adversary, the assessment scale was the same as above. Each of these factors was measured by evaluating compliance with the following corresponding statements: 'The signing of the peace agreement will strengthen democracy', 'I believe in the good intentions of demobilized ex-combatants with respect to society in general', 'I feel identified with my neighborhood or community', and 'I feel that I want to share pleasant activities or do entertaining things with people who have hurt me'.

*2.3. Data Analysis*

The data were analyzed in three phases. First, to examine the Political and Interpersonal Reconciliation Scale's factor structure we conducted a principal component analysis with varimax rotation, where factors with eigenvalues > 1 and items with factorial weights > 0.58 were selected. Second, to assess its reliability we used the alpha coefficient with the criteria of $\alpha \geq 0.70$ (Nunnally 1978). Third, to assess its external validity we performed Pearson correlations with four external related variables. For this analysis, we used R version 3.5.2 (R Foundation for Statistical Computing, Vienna, Austria).

*2.4. Ethical Requirements*

Participants were informed about the purpose of the study, the type of participation requested and the uses of the results, guaranteeing the confidentiality of the data. Then,

those interested in participating signed an informed consent form authorizing the use and disclosure of the research results.

## 3. Results

### 3.1. Structural Validity and Reliability

The analysis showed a two-factor structure with a variance explanation percentage of 73%. Bartlett's sphericity test and the Kaiser–Meyer–Olkin (KMO) measurement adequacy test yielded values of 12.557 ($p < 0.5$) and 0.50, respectively, confirming that it was appropriate to apply exploratory factor analysis to the data matrix being studied (Kaiser 1970; Tabachnick and Fidell 2001).

Table 2 presents the results of the questionnaire's factorial structure and Cronbach's alpha coefficient. The table shows statistically satisfactory values for alpha with a score of 0.86 for the two factors, indicating the questionnaire's favorable internal consistency (González and Pazmiño 2015; Oviedo and Campo 2005). Furthermore, a Cronbach's alpha of 0.84 was obtained for the entire questionnaire. Consequently, the items of political and interpersonal dimensions produced stable measurements.

**Table 2.** Factorial structure and Cronbach's alpha of the PIRS.

|  | 1<br>Political Dimension | 2<br>Interpersonal Dimension |
|---|---|---|
| PIRS-1 | 0.87 | |
| PIRS-2 | 0.87 | |
| PIRS-3 | 0.67 | |
| PIRS-4 | 0.58 | |
| PIRS-5 | 0.85 | |
| PIRS-6 | 0.81 | |
| PIRS-7 | | 0.64 |
| PIRS-8 | | 0.81 |
| PIRS-9 | | 0.81 |
| PIRS-10 | | 0.86 |
| PIRS-11 | | 0.77 |
| PIRS-12 | | 0.86 |
| Cronbach's Alpha ($\alpha$) | 0.86 | 0.86 |

Note. PIRS = Political and Interpersonal Reconciliation Scale.

### 3.2. External Validity

Table 3 presents the Pearson correlations between the Political and Interpersonal Reconciliation Scale and four external variables. In line with the hypothesis, it was positively related to the expected variables in different ways. The political dimension was positively and significantly related to confidence in the peace agreement, trust in the ex-combatants and willingness to share with the adversary. This suggests that people who feel more comfortable with the implementation of mechanisms concerning truth, justice, and reparations after a conflict tend to feel more confident about the peace agreement, as well as more willing to trust the ex-combatants and share pleasant activities with their former adversaries.

**Table 3.** Results of the correlational analysis between the PIRS and external variables.

| External Validation Criteria | Political Dimension | Interpersonal Dimension |
|---|---|---|
| Confidence in the peace agreement | 0.23 ** | 0.06 |
| Trust in the ex-combatants | 0.36 ** | 0.18 * |
| Willingness to share with the adversary | 0.28 ** | 0.17 * |
| Community identification | 0.00 | 0.40 ** |

Note. * $p < 0.05$; ** $p < 0.01$.

Meanwhile, the interpersonal dimension showed a positive and strong relationship with community identification, as well as a moderate relation with trust in the ex-combatant and willingness to share with the adversary. Therefore, people who perceive greater social support and participate in community activities, seem to feel a greater sense of identification with their community and are also open to the possibility to trust the ex-combatants and share time and pleasant activities with their former adversaries.

## 4. Discussion

The purpose of this study is to evaluate the psychometric properties of the Political and Interpersonal Reconciliation Scale. The results contributed to the literature on reconciliation by showing that political and interpersonal dimensions of reconciliation need to be measured adequately and by revealing that the scale under study exhibited statistically satisfactory reliability.

Consistent with theoretical predictions, four variables that are conceptually related to the political and interpersonal dimensions of reconciliation are closely related to the scale under study, facilitating a new measurement of reconciliation with a contingency that assumes the context that arises in a peacebuilding process. In this way, the political dimension was positively related to trust in the peace agreement, which corroborates the results of both Bloomfield et al. (2003), as well as Wolpe and McDonald (2008). Moreover, the interpersonal dimension was positively related to a willingness to trust the ex-combatant, as reported by Wagner (2006) and Alzate et al. (2009), and to community identification, as shown by Bloomfield (2006). Furthermore, both dimensions were related to a willingness to share with the adversary in a new scenario of peace. In general, it was verified that each dimension maintains different association patterns with the remaining variables of interest. This indicates the questionnaire's quality since the proposed items are capable of empirically capturing distinctions proposed at the conceptual level.

These findings are useful to researchers who are interested in the study of reconciliation in peacebuilding contexts, who may find the Political and Interpersonal Reconciliation Scale useful to have an accurate reading of reconciliation and consider the particularities of its constitutive dimensions without assigning primacy to one dimension over another.

One of the peculiarities of this study is the emphasis on the quantitative measurement of reconciliation. In fact, Rettberg (2014) stated that reconciliation has not received the academic attention it deserves because it is considered 'difficult to measure' (Rettberg 2014, p. 3). Some scales evaluate the socio-emotional and instrumental aspects of interaction in conflict societies, such as those developed by Mukashema and Mullet (2010) and Alzate et al. (2013). Other authors have developed measurement scales to meet specifically their research objectives (Kosic et al. 2011; Kosic and Livi 2012; Shnabel et al. 2009). Neither of these approaches consider reconciliation's political dimension nor the structural aspects that imply a conflict's constructive transformation, such as political and economic responses that make it possible to change a conflict's roots and 'reposition it on a path of transformation toward peace' (Fisas 2004, p. 24).

Other scales assess the attitudes toward structural mechanisms that facilitate reconciliation, such as the one proposed by Taylor (2015), or indexes that were designed to quantify the socio-political impact of reconciliation and peacebuilding projects (United Nations Office for the Coordination of Humanitarian Affairs 2014; UNDP-ACT and Centre for Sustainable Peace and Democratic Development 2015). Although this approach deals with reconciliation's political dimension, it does not consider relational aspects that converge at the interpersonal level, such as building confidence and community participation. Additionally, it does not consider that reconciliation implies direct interaction between people who need this interaction to redefine the terms of their future coexistence (Bloomfield 2006).

Concerning the Colombian case, the Political and Interpersonal Reconciliation Scale represents an opportunity to begin deconstructing and measuring reconciliation after the signing of the peace agreement, in order to have a better comprehension of the post-agreement unequivocal demand for understanding. In Colombia, the conflict began in the

19th century and was characterized by civil wars and military truces. Since it was successive and lacked successful solution processes, this conflict did not permit the specification of either reparations or reconciliation stages. Therefore, neither of the latter processes have been significantly studied in this case. Outside of constructing reconciliation scenarios, the dynamics of failed, inconclusive processes have produced actions, such as amnesty and pardon, which the civilian population was not involved in. It was only in 2016 that a peacebuilding process began with the signing of a peace agreement to end the conflict with the FARC-EP, which assumed the legal frameworks that were in force at the international and national levels. This demonstrated the need to not consider reconciliation as a process for the victims but as a national project and a structural, fundamental part of the post-agreement unequivocal demand for understanding. This also presents an opportunity to begin deconstructing, measuring, and defining reconciliation, as it is proposed in this study.

In Colombia, this study represents an opportunity to continue understanding and building reconciliation, considering that even if the conflict with the FARC-EP has ended, this society continues to present a high level of social conflict, due to the presence of other insurgent groups in the territory, organized crime gangs, and clashes over land among ethnic minorities such as Afro-descendants and indigenous people.

## 5. Conclusions

The Political and Interpersonal Reconciliation Scale measures reconciliation in a way that equally integrates its political and interpersonal dimensions while simultaneously guaranteeing reliability criteria. It broadens the understanding of reconciliation in peace-building processes and facilitates a comprehensive approach to researchers that aim to identify reconciliation's predictors, as well as the comparison of research results in different contexts.

In relation to the sample, there is a prevalence of young, middle-class participants located on the northern coast of Colombia. Despite these limitations, this study contributes to the study of political and interpersonal reconciliation through the design and validation of an instrument for its evaluation in a local context. The aim is to move towards the implementation of more experimental studies in order to carry out interventions to foster reconciliation. For future research, there is a need to replicate the present study in other geographic contexts, with different populations, including representative samples of different ages who identify themselves as victims as well as ex-combatants and compare the results. In addition, it would be useful to compare these findings with other relevant findings in cross-cultural settings. Particularly in the case of Colombia, it would also be appropriate to distinguish the differences between rural and urban populations, since rural and urban areas experienced the conflict in a unique way, and it is expected to be the same regarding their post-agreement experience. Authors should discuss the results and how they can be interpreted from the perspective of previous studies and the working hypotheses. The findings and their implications should be discussed in the broadest context possible. Future research directions may also be highlighted.

**Author Contributions:** Conceptualization, E.S.C. and I.M.-G.; methodology, E.S.C. and I.M.-G.; formal analysis, E.S.C., I.M.-G., R.G.A.; investigation, E.S.C., M.M.P.; resources, E.S.C. and M.M.P.; data curation, E.S.C.; writing—original draft preparation, E.S.C. and I.M.-G.; writing—review and editing, E.S.C., I.M.-G. and R.G.A.; project administration, E.S.C.; funding acquisition, E.S.C. All authors have read and agreed to the published version of the manuscript.

**Funding:** This research was funded by the Fundación Universidad del Norte, grant number 2018-14. This study is part of the project 'Connected Worlds: The Caribbean, Origin of Modern World'. This project has received funding from the European Union's Horizon 2020 research and innovation programme under the Marie Skłodowska Curie grant agreement N° 823846. This project is directed by Professor Consuelo Naranjo Orovio, Instituto de Historia-CSIC.

**Institutional Review Board Statement:** The study was conducted in accordance with the Declaration of Helsinki and approved by the Ethics Committee of FUNDACIÓN UNIVERSIDAD DEL NORTE (protocol code N°. 194 and approved on 22 August 2019).

**Informed Consent Statement:** Informed consent was obtained from all subjects involved in the study.

**Data Availability Statement:** Not applicable.

**Conflicts of Interest:** The authors declare no conflict of interest.

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
