# Peer review of "A Quantitative Exploration of Reconciliation: Evidence from Colombia"

_socsci, doi:10.3390/socsci11100456_

Round 1

Reviewer 1 Report

The authors collected original data from Colombia to build a scale that measures the interpersonal dimension of reconciliation to complement the existing scale that measures the political dimension of reconciliation.

The research topic is critical that I believe will eventually be added to the peace science literature. However, I have two issues with the piece.

First, the authors do not convincingly explain the necessity of measuring psychometric properties. Perhaps the authors can approach the framing of the project from a practical point of view. For example, to better frame the project, the authors may cite historical instances of the reconciliation process where policymakers would have benefited from a psychometric scale. The authors may also illustrate how policymakers seeking to implement a policy to facilitate and expedite the reconciliation process may benefit from the new psychometric scale that measures the stakeholders’ readiness.  

Second, the way the authors prove validity comes across as truistic; they conclude that the developed scale is valid because the new variables are highly correlated with the existing metric. It is true that the Principal Component Analysis (PCA) can be used to test the internal validity and reliability of a questionnaire when responses should provide a coherent and consistent narrative. However, in this case, the assumption should not be that the political measure, which is the existing scale, and the psychometric measure that the authors newly developed are consistent. In fact, if the new measure is to be useful, it should provide information that the existing one cannot capture and therefore the validity of the new measure should not depend on the similarity (correlation) to the existing one.

Some minor issues:

I suggest that the authors include a paragraph detailing the data collection process. What was the sampling strategy? How were the 171 Colombian civilians sampled and contacted? What could be the possible sources of selection biases? How do the authors address them?

Finally, the piece would be more compelling if the authors focused more on specific aspects currently being measured by discussing the Political and Interpersonal Reconciliation Scale more fully.

Reviewer 2 Report

An excellent article in a field where more research is needed. Very good is the distincton between socio-political and interpersonal reconciliation. A few more recent works could and should be quoted.

Reviewer 3 Report

Overall, I think this paper is an original and interesting contribution to the literature, and the author(s) deserve credit for their innovative survey. I think the manuscript is, rightly, likely to be accepted. I am not familiar with the specific case (Columbia) in question, but I know that in many civil wars, they primarily occur in specific regions of the country, making them a highly salient issue for those in the surrounding territory, but not at all salient elsewhere in the country (see, for example, the long-standing civil war in the Democratic Republic of Congo).  In terms of the survey sample, the author(s) do an adequate job ensuring they sample people with multiple perspectives, however,  I did not see a mention of where the respondents were located.  Additionally, the sample seems to skew young, and in recent years the violence has been less intense (this is demonstrated by well over half of the respondents not identifying themselves as victims of the conflict). The only place this is addressed is in the conclusion.

Thus, my primary critique centers on the lack of attention geography and time are given in the manuscript, as well as the youth in the sample. The author(s )should make more of an attempt to address those potentially very important and clearly omitted factors. In my reading, this falls somewhere between a decision of "accept with minor revision" and "reconsider after major revision". If the author(s) are unable to do something to account for those indicators, they should clearly acknowledge their missingness and the limitations that missingness places on their contribution to the literature.

Reviewer 4 Report

This is a good article and I hope to see it published soon. I was surprised not to see any mention of various Reconciliation Barometers created around the world (even in Colombia) to try to measure reconciliation. Even if short a commentary of a comparison between the Political and Interpersonal Reconciliation Scale and other barometers is needed before publication. This I am sure will strengthen and make clear the originality and contribution to scholarship.

Round 2

Reviewer 3 Report

I find the authors response sufficiently engaging and persuasive and recommend acceptance